# Social support and symptoms of antenatal depression among women screened for gestational diabetes mellitus: A cross-sectional study in Northern Vietnam (the VALID II study)

**Ai T. Nguyen**[1]*, **Kien Dang Nguyen**[2], **Hieu Minh Le**[3], **Thanh D. Nguyen**[1], **Dan W. Meyrowitsch**[4], **Ib C. Bygbjerg**[4], **Jens Søndergaard**[5], **Hanh T. T. Nguyen**[6], **Christina A. Vinter**[7,8,9], **Ditte S. Linde**[7,8], **Tine M. Gammeltoft**[10], **Vibeke Rasch**[7,8]

1 Faculty of Public Health, Thai Binh University of Medicine and Pharmacy, Thai Binh, Vietnam, 2 Department of Obstetrics & Gynecology, Thai Binh University of Medicine and Pharmacy, Thai Binh, Vietnam, 3 Department of Internal Medicine, Thai Binh University of Medicine and Pharmacy, Thai Binh, Vietnam, 4 Department of Public Health, Global Health Section, University of Copenhagen, Copenhagen, Denmark, 5 Department of Public Health, Research Unit of General Practice, University of Southern Denmark, Odense, Denmark, 6 Department of Demography, Ha Noi Medical University, Ha Noi, Vietnam, 7 Department of Gynaecology and Obstetrics, Odense University Hospital, Odense, Denmark, 8 Department of Clinical Research, University of Southern Denmark, Odense, Denmark, 9 Steno Diabetes Center Odense, Odense University Hospital, Odense, Denmark, 10 Department of Anthropology, University of Copenhagen, Copenhagen, Denmark

* nguyenai198@gmail.com

## Abstract

### Objectives

This study from Northern Vietnam aims to assess the association between social support and symptoms of depression among pregnant women screened for gestational diabetes mellitus (GDM).

### Methods

A cross-sectional study was conducted among 823 pregnant women in Thai Binh, Vietnam. The women were screened for GDM and structured questionnaire were used to collect data on social support factors, GDM factors, and symptoms of depression. The diagnosis of GDM was based on the 2-hour 75-g OGTT according to WHO 2013 criteria. The Edinburg Postpartum Depression Scale (EPDS) with a cut-off of 10 and the Multidimensional Perceived Social Support Scale (MSPSS) were used to assess depression symptoms and perceived social support, respectively. Logistic regression analysis was conducted to measure the associations between social support, GDM-related factors, and symptoms of depression. The relationship between social support score and symptoms of depression was evaluated using Spearman's correlation. The strength of the associations were measured by adjusted odds ratios (aOR) with 95% confidence intervals (CI).

**Data Availability Statement:** The data for this study is available at the DOI 10.6084/m9.figshare. 27794355 or through the link https://figshare.com/ articles/dataset/Data_depression_revised_xlsx/ 27794355.

**Funding:** this study was founded by the Danish Ministry of Foreign Affairs. the funder had no role in study design, data collection and analysis, decision to publish, or preparation of the manuscript.

**Competing interests:** The authors have declared that no competing interests exist.

## Results

The prevalence rates of GDM and symptoms of depression were 22.2% (95%CI: 19.4–25.2) and 23.0% (95%CI: 20.1–26.0), respectively. Women who had moved away from their commune of birth and women who reported another person than their husband to be the primary person to confide in had increased odds of depression (aOR = 1.74; 95%CI:1.19–2.56 and aOR = 2.36; 95%CI:1.48–3.75, respectively). A reported lack of social support was strongly associated with increased odds of depression symptoms among both women with gestational diabetes mellitus (aOR = 6.16, 95% CI:2.35–16.12) and without gestational diabetes mellitus (aOR = 2.81; 95%CI: 1.67–4.75). When analysing the correlation between social support and depression symptoms, a negative correlation was found, with decreasing depression scores as the social support score increased.

## Conclusion

The prevalence of symptoms of depression was high in our study, and women in Northern Vietnam who feel well-supported socially are less likely to report symptoms of depression. This finding applies both to women with and without GDM.

## Introduction

Pregnancy is a period of increased vulnerability due to the physiological and psychosocial changes women experience while being pregnant. Consequently, the prevalence of anxiety and depression is as high as 15–65% in pregnant women, with rates beinghigher in low- and middle-income countries where a prevalence of 25.5% has been reported in a recent meta-analysis [1, 2]. Recent attention has been drawn to depressive symptoms in pregnant women with gestational diabetes mellitus (GDM), as studies suggest a link between GDM, increased psychological stress, and depression [3, 4]. In Asian countries, the prevalence of GDM is increasing, with prevalence rates of 24.7% in Thailand, 23.5% in Singapore, 22.5% in Malaysia, and 21.3% in Vietnam [5]. Depression signs in pregnancy is regarded as common and serious problem due to its chronic nature and potential for recurrence. Moreover, it can have significant adverse effects on the health of both the woman and the fetus [6, 7].

Depression in pregnant women is influenced by various factors, including history of abortion, previous obstetric complications, comorbidities, experiences of partner violence, negative events during pregnancy, and lack of social support [8, 9]. Additionally, changes in life sitiation can affect the incidence of depression [9]. There is also evidence of a strong correlation between increased blood glucose levels and depression [10–12], possibly due to the development of brain insulin resistance [13]. Further, extensive evidence shows that pregnant women without adequate social support are at higher risk of developing mental illnesses compared to those with strong social support [14–16].

Social support is defined as the perception or experience that one is loved or cared for by others such as friends, coworkers, family members, partners, as well as community members [17]. In the context of GDM, there has been speculation about whether a lack of social support contributes to difficulties in managing healthy lifestyles, potentially increasing the risk of GDM. Consequently, social support may be linked to better blood glucose control and fewer adverse obstetric outcomes among women with GDM [18, 19]. However, there is limited data from low and middle-income countries regarding the associations between social support,

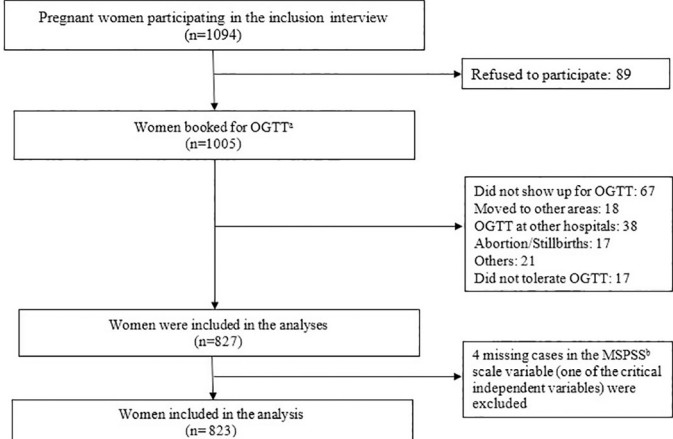

**Fig 1. Flowchart of participants in the study.** [a]: Oral glucose tolerance test [b]: Multidimensional Perceived Social Support Scale.

mental health, and GDM and to the best of our knowledge, no such studies have been conducted in Vietnam. Therefore, this study aims to assess the association between social support and symptoms of depression among pregnant women who were screened for GDM. Furthermore, it aims to measure the associations between levels of social support and signs of depression when stratified by women with and without GDM.

## Materials and methods

### Study setting

We conducted a cross-sectional study nested within the ongoing "Living Together with Chronic Disease: Informal Support for Diabetes Management in Vietnam, Phase II: Gestational Diabetes in Vietnam (VALID II)". Thai Binh Province covers an area of 1,542 km$^2$ and has approximately 1,860,000 inhabitants living in seven rural districts and one capital city, Thai Binh City, with an approximate population of 200,000. Data were collected at Thai Binh Maternity Hospital and Kim Ngan Clinic in Thai Binh City. Thai Binh Maternity Hospital, with approximately 11.000 annual deliveries, is the only governmental maternity hospital in the Thai Binh Province. Kim Ngan Clinic is a private clinic where women can receive antenatal care with approximately 8.000 cases per year.

### Population

In total, 1094 pregnant women attending antenatal care were consecutively invited to participate in the study and offered an oral glucose tolerance test (OGTT) in gestational age (GA) 24–28 weeks. Inclusion criteria were: (1) Pregnancy ≤ 28 weeks; (2) Singleton and multiple pregnancies; (3) Residing in Thai Binh Province; (4) Speaking and reading Vietnamese; and (5) Agreeing to participate in the study after informed consent. Exclusion criteria were: 1) Pregestational diabetes (type 1 or type 2); and 2) Acute episodes of chronic diseases such as: kidney disease, heart failure, cancer . . . (Fig 1).

### Data collection

Data were collected from January 2023 to August 2023. Face to face interviews were conducted by trained nurses who received thorough training from the research team at Thai Binh

University of Medicine and Pharmacy regarding the research objectives, the research questionnaire, approaches to the study participants, interview techniques, situation handling, and maintaining communication with the women. Women were invited to participate in the study when they attended antenatal care in 5–28 gestational week. If the woman consented to participate in the study, she participated in a short inclusion questionnaire interview with the study nurse/midwife and was scheduled for an OGTT in GA from 24–28 weeks. The OGTT interview was conducted while the pregnant woman was waiting for the OGTT results.

## Measurements

The inclusion interviews focused on socioeconomic characteristics (age, place of residence, occupation, marital status, and number of children). The OGTT interview concerned family relations, social support, depression symptoms, and GDM-related risk factors (physical activity level and pregestational BMI). These data were collected while waiting for the outcome of the OGTT, i.e., the GDM status was not known when social support and symptoms of depression were measured.

The Multidimensional Perceived Social Support Scale (MSPSS) [20] was used to assess the women's perceived social support. It includes 12 items that cover three dimensions of social support: family, friends, and significant others. In addition, the total score is divided into three levels: *low* perceived support (12–35 score), *medium* perceived support (36–60), and *high* perceived support (61–84). Each item is rated on a seven-point Likert scale from 1–7 (min-max scores: 12 and 84). The higher the score, the higher the perceived social support. For each dimension (family, friends, and significant others), the minimum-maximum score ranged from 4–28. In the present study, each of the three dimensions of perceived support was dichotomized into low and high levels, using a cut-off point of 75% of the total score. Thus, scores ranging from 4–21 and 22–28 indicated low and high levels, respectively. Cronbach's alpha was used to measure the internal consistency of the 12 questions in the MSPSS toolkit, and the results showed an alpha coefficient of 0.91. Depression symptoms were assessed using the Edinburgh Postnatal Depression Scale (EPDS) [21], a 10-item validated screening tool aiming at detecting signs of postnatal depression. The tool has been validated for use during pregnancy [22, 23] and according to a systematic review, the EPDS can be used to assess depressive disorders among pregnant women [24],. Women rated how often they experienced certain feelings or behaviors on a 4-point Likert scale ranging from 0–3 (min-max score: 0–30). A cutoff of 10 or above was used to identify women with symptoms of depression in line with previous research from Vietnam [25]. The EPDS toolkit's internal consistency, the ten questions' alpha coefficient was 0.81.

The GDM-related factors included physical activity and BMI. Physical activity was measured as hours spent on low-intensity (not sweating/out of breath) and high-intensity (sweating/out of breath) physical activity per week [26]. BMI was assessed using the Asian cut-off points for underweight (BMI<18.5kg/m$^2$), normal weight (BMI from 18.5 to 22.9 kg/m$^2$), overweight (BMI from 23.0 to 25.99 kg/m$^2$), and obesity (BMI $\geq$25.0 kg/m$^2$) [27]. A diagnosis of GDM was based on the 2-hour 75-g OGTT. The following criteria were used to denote a positive test result: Fasting plasma glucose $\geq$ 5.1–6.9mmol/L; 1-hour glucose $\geq$ 10mmol/L; 2-hour glucose $\geq$ 8.5–11.0 mmol/L according to the WHO 2013 criteria [28].

## Data management and analysis

Data were entered into REDCap and exported to the Statistical Package for Social Sciences (SPSS window version 22) for analysis. Descriptive statistics, including frequencies,

percentages, means and standard deviations (SD), and median and interquartile range (IQR), were utilized to summarize the women's sociodemographic characteristics.

Logistic regression analysis was conducted to measure the associations between social support, GDM-related factors, and symptoms of depression. Stratified analyses of the association between the three MSPSS dimensions and symptoms of depression among women with and without GDM were additionally performed. The strength of the association was measured by odds ratios (OR) with 95% confidence intervals (CI). A cut-off of $p<0.05$ was used for statistical significance. Subsequently, we adjusted for socio-demographic factors (age, marital status, living area, education, occupation, economic status), GDM-related factors (pregestational BMI, physical activity), and social support-related factors (place of birth and entrusted person). We selected these factors a priori based on the literature and their availability in the dataset [29, 30]. Finally, the relationship between social support score (MSPSS scale) and symptoms of depression was evaluated using Spearman's correlation. All questionnaires were managed and stored securely at the Thai Binh University of Medicine and Pharmacy (TBUMP).

### Ethics and trial registration

The study was approved by the Ethics Council in Biomedical Research of TBUMP, Vietnam (No. 1325/HDDD, 1st December 2022). Informed written consent was obtained, and it was stressed that the participants could withdraw their consent at any time during the study. Both oral and written informed consent were obtained from all research participants and confidentiality was guaranteed. All methods were carried out in accordance with relevant guidelines and regulations. The protocol for the overall VALID-II study was registered at clintrial.gov (NCT05744856). Women who reported any thoughts of suicidal intentions (question E10), were referred for further counseling by a psychologist.

## Results

In all 1005 pregnant women accepted participation in the study and were booked for OGTT and test results were available among 823 of these women. A total of 183 women (22.2%; CI95%: 19.4–25.2) were diagnosed with GDM, and 189 women (23.0%; CI95%: 20.1–26.0) had EPDS score of 10 or above, indicating symptoms of depression. The characteristics of the participants are presented in Table 1, the average age of the women was 28.3±5.3 years, with the majority aged 25–29 (35.4%). More than half were living in an urban area (59.3%), had attended college or university (58.9%), and were workers/farmers (32.4%). In addition, most women lived with their husbands (93.1%) and considered their economic status medium (98.3%). Almost half of the women were first-time pregnant (42.3%), more than one-fourth had GDM (28.3%), and 16% had a pregestational BMI $\geq$ 23kg/m$^2$.

Table 2 summarises the association between family relations, GDM-related factors, and symptoms of depression. After adjusting for age, marital status, living area, education, occupation, and economic status, higher odds of having symptoms of depression were observed among women who reported they had moved away from their home commune (aOR = 1.74; 95% CI:1.19–2.56) and among those who reported another person than their husband to be the primary person they confided in if experiencing troubles (aOR = 2.36; 95% CI:1.48–3.75). Additionally, a positive association was found between low MSPSS mean scores and symptoms of depression ($p<0.0001$). No association was found between GDM-related factors such as BMI, physical activity, blood sugar level, and symptoms of depression.

In general, a high level of social support was reported by the majority of women (86.8–87.1%), while only a few (12.9–13.2%) reported it to be medium and none (0%) to be low (Table 3). There was a tendency for women with GDM, who reported a medium level of social

**Table 1. Maternal baseline characteristics (n = 823).**

| Characteristics | No. of women (% of total) |
|---|---|
| **Age (Mean, SD)** | 28.3 (5.3) |
| **Age, years** | |
| < 25 | 240 (29.2) |
| 25–29 | 292 (35.4) |
| 30–34 | 196 (23.8) |
| 35–39 | 73 (8.9) |
| ≥ 40 | 22 (2.7) |
| **Living areas** | |
| Rural | 335 (40.7) |
| Urban | 488 (59.3) |
| **Highest completed educational level** | |
| Primary school | 2 (0.2) |
| Secondary school | 77 (9.4) |
| High school | 259 (31.5) |
| College/university and above | 485 (58.9) |
| **Occupation** | |
| Civil servant | 248 (30.1) |
| Worker, farmer | 267 (32.4) |
| Free labour | 114 (13.9) |
| Small trade | 90 (10.9) |
| Unemployed /Student/Housewife/No longer in job due to pregnancy | 97 (11.8) |
| Others | 7 (0.9) |
| **Marital status** | |
| Living with their husbands | 766 (93.1) |
| Not living with their husbands | 46 (5.6) |
| Separated, divorced, others | 11 (1.3) |
| **Number of living children in the household** | |
| 0 | 348 (42.3) |
| 1 | 309 (37.5) |
| 2 | 137 (16.7) |
| ≥ 3 | 29 (3.5) |
| **Economic status (self-reported)** | |
| Poor, near-poor | 4 (0.5) |
| Medium | 809 (98.3) |
| Wealthy | 10 (1.2) |
| **Gestational diabetes mellites** | |
| Yes | 233 (28.3) |
| No | 590 (71.7) |
| **Pregestational BMI\*, kg/m$^2$** | |
| Underweight (BMI <18.5) | 159 (19.3) |
| Normal (BMI from 18.5 to 22.9) | 532 (64.7) |
| Overweight (BMI from 23.0 to 24.99) | 86 (10.4) |
| Obesity (BMI ≥25) | 46 (5.6) |
| **Physical activity, hours/week** | |
| low-intensity physical activity (Mean, sd) | 5.3 (12.3) |
| high-intensity physical activity (Mean, sd) | 0.2 (2.2) |
| **Symptoms of depression (EPDS score ≥10)** | |

*(Continued)*

**Table 1.** (Continued)

| Characteristics | No. of women (% of total) |
|---|---|
| Yes | 189 (23.0) |
| No | 634 (77.0) |
| **Multidimensional Perceived Social Support Scale (MSPSS)** | |
| Low | 0 (0.0) |
| Medium | 108 (13.1) |
| High | 715 (86.9) |

\* **BMI**: Body mass index

support, to have increased odds of symptoms of depression (aOR 6.40; 95% CI 2.44–16.83) in comparison to women who reported a high level of social support (aOR 2.81; 95%:1.67–4.75). When focusing on different dimensions of social support, women with lower family support scores had higher odds of having symptoms. Among women with low family support scores compared to those with high family support scores, the odds of symptoms of depression were higher among women with GDM (aOR = 4.51; 95%CI: 1.85–10.97) compared to those without GDM (aOR = 3.31; 95%CI: 1.94–5.64). Furthermore, lack of friend support was associated with increased odds of depression symptoms, and the association also here tended to be more pronounced among women with GDM (aOR = 2.85; 95% CI: 1.37–5.95) in comparison with those without GDM (aOR = 1.90; 95% CI: 1.26–2.87). The trend was similar among women who reported support from significant others, where aORs were 3.04 (95% CI: 1.31–7.48) among women with GDM and 2.02 (95% CI:1.22–3.34) among those without GDM.

Table 4 provides an overview of the correlation between MSPSS and EPDS scores among women with and without GDM. A negative correlation between EPDS score and the scores in social support from family, friends, and others was found among both women with GDM and women without GDM. Accordingly, there was a trend to be an increase in depressive symptoms when social support decreased.

## Discussion

Almost one-fourth of the women in our study were diagnosed with GDM and 23% reported symptoms of depression. Women who lived in the same area as they were born and women who entrusted their husbands as primary confidants were less likely to report symptoms of depression. Also, women who stated they had social support from family, friends, and significant others were less likely to have symptoms of depression.

### Interpretation

The prevalence of symptoms of depression was high in our study (23.0%). Similar high depression rates have been reported in other Vietnamese studies. E.g: a study in 2020 from Hanoi, Thanh Hoa, Hue, and Ho Chi Minh, which used a similar EPDS cut-off point, found that 24.5% of pregnant women had symptoms of depression [29]. However, in contrast with our findings, another study from Northern Vietnam in 2019, using a cut-off of 10 only found a depression symptom rate of 5% in pregnant women [30]. Internationally, there are also great variations in rates of depression symptoms, with reported prevalence rates of 8.6% in Western Europe, 7.0% in Australia, 17.9% in Northeast Ethiopia, and 14.8% in Brazil [31–35]. The differences may be due to variations in cutoff points, study time and durations, geographical locations of the research, and cultural differences across regions. Therefore, future studies should

**Table 2. Association between family relations, GDM related factors, and symptoms of depression (n = 823).**

| Characteristics | EPDS score <10 (n = 634) N (%) Or mean, sd | EPDS score ≥10 (n = 189) N (%) Or mean, sd | Crude odds ratio (CI95%) | Model 1 aOR (95% CI) [a] | Model 2 aOR (95% CI) [b] |
|---|---|---|---|---|---|
| **Family relations** | | | | | |
| **Current residence of women (n = 822)** | | | | | |
| Not in the same commune as place of birth | 411 (74.6) | 140 (25.4) | **1.54 (1.07–2.22)** | **1.76 (1.20–2.58)** | **1.74 (1.19–2.56)** |
| The same commune as a place of birth | 222 (81.9) | 49 (18.1) | **1** | **1** | **1** |
| **See and talk with own parents (n = 817)** | | | | | |
| No | 102 (79.1) | 27 (20.9) | 0.87 (0.55–1.38) | 0.74 (0.46–1.20) | 0.73 (0.45–1.19) |
| Yes | 528 (76.7) | 160 (23.3) | 1 | 1 | 1 |
| **Count on members of family of birth for support (n = 822)** | | | | | |
| No | 21 (70.0) | 9 (30.0) | 1.45 (0.66–3.24) | 1.12 (0.47–2.72) | 1.07 (0.44–2.64) |
| Yes | 612 (77.3) | 180 (22.7) | 1 | 1 | 1 |
| **Count on members of the husband's family for support (n = 817)** | | | | | |
| No | 33 (68.7) | 15 (31.3) | 1.57 (0.83–2.95) | 1.45 (0.72–2.92) | 1.46 (0.72–2.95) |
| Yes | 596 (77.5) | 173 (22.5) | 1 | 1 | 1 |
| **When in trouble, counting on (n = 823)** | | | | | |
| Others | 62 (62.6) | 37 (37.4) | **2.25 (1.44–3.50)** | **2.38 (1.50–3.79)** | **2.36 (1.48–3.75)** |
| Husband | 572 (79.0) | 152 (21.0) | **1** | **1** | **1** |
| **GDM related factors** | | | | | |
| **Pregestational BMI (n = 815)** | | | | | |
| Overweight, obesity | 101 (76.5) | 31 (23.5) | 1.04 (0.67–1.61) | 1.05 (0.66–1.66) | 1.06 (0.67–1.68) |
| Underweight, normal | 533 (77.1) | 158 (22.9) | 1 | 1 | 1 |
| **Gestational diabetes mellitus (n = 823)** | | | | | |
| Yes | 183 (78.5) | 50 (21.5) | 0.89 (0.62–1.28) | 0.86 (0.58–1.27) | 0.85 (0.57–1.25) |
| No | 451 (76.4) | 138 (23.6) | 1 | 1 | 1 |
| **OGTT result (n = 823)** | | | | | |
| Fasting plasma glucose (mmol/l) (mean, sd) | 4.6 (0.4) | 4.6 (0.4) | 0.801* | | |
| 1-hour plasma glucose (mmol/l) (mean, sd) | 8.0 (1.8) | 8.1 (1.7) | 0.748* | | |
| 2-hour plasma glucose (mmol/l) (mean, sd) | 7.2 (1.4) | 7.3 (1.3) | 0.518* | | |
| **Physical activity (n = 799)** | | | | | |
| low-intensity physical activity (mean, sd) | 5.2 (12.5) | 5.5 (11.9) | 0.43* | | |
| high-intensity physical activity (mean, sd) | 0.2 (2.3) | 0.2 (1.8) | 0.431* | | |

* Mann-Whitney U test

GDM: Gestational diabetes mellites; EPDS: Edinburgh Postnatal Depression Scale; MSPSS: Multidimensional Perceived Social Support Scale; OGTT: Oral glucose tolerance test

[a] Adjusted for age, marital status, living area, education

[b] Adjusted for age, marital status, living area, education, occupation, economic status

**Table 3. Perceived social support and depressive symptoms in women with and without gestational diabetis mellitus (GDM).**

| Factors | GDM N = 233 | | | | No GDM N = 590 | | | |
|---|---|---|---|---|---|---|---|---|
| | EPDS score < 10 (n = 183) | EPDS score ≥ 10 (n = 50) | OR (95% CI) | aOR (95% CI) | EPDS score < 10 (n = 451) | EPDS score ≥ 10 (n = 139) | OR (95% CI) | aOR (95% CI) |
| **MSPSS scale (total score)** * | | | | | | | | |
| High perceived support (61–84) | 168 (82.8) | 35 (17.2) | 1 | 1 | 407 (79.5) | 105 (20.5) | 1 | 1 |
| Medium perceived support (36–60) | 15 (50.0) | 15 (50.0) | 4.80 (2.15–10.72) | 6.40 (2.44–16.83) | 44 (56.4) | 34 (43.6) | 2.78 (1.65–4.68) | 2.81 (1.67–4.75) |
| Low perceived Support (0–35) | 0 | 0 | NA | NA | 0 | 0 | NA | NA |
| **MSPSS subscale** * | | | | | | | | |
| *Family subscale* | | | | | | | | |
| Sub-score 22–28 | 165 (82.5) | 35 (17.5) | 1 | 1 | 412 (79.5) | 106 (20.5) | 1 | 1 |
| Sub-score 4–21 | 18 (54.5) | 15 (45.5) | 3.93 (1.81–8.54) | 4.51 (1.85–10.97) | 39 (54.2) | 33 (45.8) | 3.28 (1.92–5.60) | 3.31 (1.94–5.64) |
| *Friends subscale* | | | | | | | | |
| Sub-score 22–28 | 103 (85.1) | 18 (14.9) | 1 | 1 | 248 (82.4) | 53 (17.6) | 1 | 1 |
| Sub-score 4–21 | 80 (71.4) | 32 (28.6) | 2.29 (1.20–4.37) | 2.85 (1.37–5.95) | 203 (70.2) | 86 (29.8) | 1.90 (1.26–2.87) | 1.92 (1.29–3.0) |
| *Others subscale* | | | | | | | | |
| Sub-score 22–28 | 157 (81.8) | 35 (18.2) | 1 | 1 | 394 (78.8) | 106 (21.2) | 1 | 1 |
| Sub-score 4–21 | 26 (63.4) | 15 (36.6) | 2.59 (1.24–5.40) | 3.04 (1.28–7.22) | 57 (63.3) | 33 (36.7) | 2.02 (1.22–3.34) | 2.03 (1.23–3.36) |

* Adjusted for age, marital status, living area, education, occupation, economic status, pregestational boday mass index, physical activity, place of birth, and entrusted person

investigate the appropriate cut-off score for depression symptoms among pregnant women and related clinical outcomes within the Vietnamese context.

Like other studies, we found that social support was a protective factor against symptoms of depression [36]. Our study was conducted among pregnant women, and it may be argued that they differ from other demographic groups since social support protects the mother-to-be and yields advantages for her offspring. Hence, if psychological distress during pregnancy is left untreated, it may influence antenatal care and place the women at increased risk for miscarriage, preterm birth, and low birth weight infants [37, 38]. Moreover, untreated symptoms of

**Table 4. Correlation between social support and symptoms of depression among pregnant women screened for gestational diabetes mellitus (GDM).**

| MSPSS scale | EPDS scale | |
|---|---|---|
| | GDM | No GDM |
| Family subscale ($r_s$) | -0.18* | -0.25* |
| Friends subscale ($r_s$) | -0.19* | -0.21* |
| Others subscale ($r_s$) | -0.18* | -0.25* |
| Sum MSPSS ($r_s$) | -0.23* | -0.28* |

$r_s$: Spearman's linear correlation coefficient.
*P<0.01

depression during pregnancy may lead to postpartum depression, which may have a range of consequences for both the mother and the child [39].

In this study, we focused on how family ties were associated with symptoms of depression in pregnant women. To assess this mechanism, women were questioned about how they connected with their families and who they confided in. Women who lived in their community of birth and women who reported that their husband was the first person they confided in were significantly less likely to show symptoms of depression. These findings are supported by a systematic review showing that family support is a protective factor against depression among adults [36]. As kinship in northern Vietnam is patrilineal and patrilocal, most women leave their natal family after marriage to live with their husbands' families. In addition, traditional gender roles and cultural norms have historically positioned Vietnamese women in subordinate positions within the family, and when moving away from her natal family, the woman is often dependent on support from her husband [40]. These circumstances may help explain why women who had moved away from their native commune and women who were not able to confide in their husbands more often had symptoms of depression.

A bidirectional association between diabetes and depression has been suggested. Several studies have demonstrated that individuals with depression, anxiety, or stress-related disorders have high blood glucose levels [41–44]. The mechanistic links between the association are still under study, but it has been put forward that an abnormal glucose metabolism may partly disrupt the regulation of the hypothalamic-pituitary-adrenal axis, leading to elevated cortisol levels and subsequent depression [45–47]. Recent findings have also revealed a relationship between depression and GDM where depression, through its associated sedative behavior, may affect blood sugar control and thus lead toGDM [48, 49]. Our study did, however, not demonstrate a correlation between blood glucose level, GDM, and symptoms of depression.

It has also been suggested that physical exercise can be beneficial in treating depression as exercise can contribute to improving mood and aiding the recovery process for depressed individuals [50]. Regarding the women's level of physical activity in our study, women spent an average of 5,3 hrs per week on low-intensity physical activity and 0.2 hrs per week on high-intensity physical activity and we did not find a correlation between physical activity and depressive symptoms. The lack of association between physical activity and depression symptoms likely reflects that the physical activity reported in our study mainly concerned physical activity performed as part of the women's daily chores and not bodily activity performed to maintain physical fitness and overall health. This is in contrast to other studies where physical exercises have been used as an add-on treatment for individuals diagnosed with depression [51].

We found a strong association between overall social support and symptoms of depression with poor social support being associated with symptoms of depression among women with and without GDM. This is not surprising, yet the association appeared to be stronger among women having GDM, although the confidence interval after adjustment was quite broad due to the relatively low number of women with symptoms of depression in the different subgroups. This finding also reinforces statements about the lack of social support contributing to difficulties in managing a healthy lifestyle in pregnant women, which may increase the risk of GDM [17]. Particularly in the context of pregnant women with comorbidities who have increased healthcare needs [52, 53], lack of social support may cause poor blood sugar control in pregnant women with GDM. Hence, the women with GDM in our study may be a particularly representative group of vulnerable women who need a strong social support network and, thus, likely will benefit from social support interventions. We also assessed the different dimensions of social support and found that poor support from family was strongly related to increased odds of depression symptoms. Also, poor support from friends and significant

others was associated with increased odds of depression, although to a lower extent. Our findings are in line with other studies documenting that the most important social support person among adults suffering from depression is their spouse and family [36, 54]. However, an alternative interpretation regarding causality is that a low level of social support may not directly result in a higher risk of GDM. Rather, it could serve as a proxy indicator for families with higher levels of direct risk factors for GDM, such as an unhealthy diet and lower levels of physical activity.

Based on our findings, the involvement of the husband and family suggest a potential causal role in relation to depression in pregnant women. Specifically, as mentioned above, low social support could be regarded as a proxy for a spousal relationship or a family context that is less than optimal. This could suggest avenues for further research focusing on marital relationships and family dynamics in relation to mental health during pregnancy.

## Strengths and limitations

A strength of the present study is that the diagnosis of GDM was obtained using rigorous diagnostic criteria, with OGTT being offered to pregnant women who attended routine care and utilized antenatal care at both private and public health facilities. This approach adds to the representativity of the study population. In addition, screening for depression symptoms was performed before the women knew the OGTT results, so it was not affected by the diagnosis of GDM. Another strength is the fairly large sample size, which allowed us to examine in detail how social support is associated with depression symptoms among women with and without GDM. However, the study also has some limitations. Firstly, the cross-sectional nature of the analyses makes it challenging to conclude the direction of causality between social support and depression symptoms. Secondly, depression and self-perception of social support for oneself are sensitive topics, and therefore, some women may not have disclosed their experiences. Underestimation of the number of women with signs of depression would most likely lead to an underestimation of the true associations between social support and symptoms of depression [55, 56]. Previous research in Vietnam has shown that women tend to minimize or conceal their feelings of stress partly to avoid burdening others and partly due to community stigma surrounding mental health [55–57]. Furthermore, results from a previous study indicate that pregnant women with gestational diabetes tend to prioritize the health of their children over their own [58]. Finally, the MSPSS scale relies on self-reported information on social support and it has been validated as an appropriate tool for identifying perceived social support in pregnant women [59]. The self-reporting of social support levels may be biased as it could be a sensitive issue. This may explain why none of the pregnant women in our study reported receiving low social support.

## Conclusion

Depression symptoms are common in pregnant women in Northern Vietnam and limited social support, especially poor social support from husband and family, is strongly associated with an increased risk for sign of depression. This applies to both women with GDM and women without GDM. There is a need to acknowledge and address the risk of depression in pregnant women. This requires a multifaceted approach, which includes raising awareness among professionals, promoting self-care practices, and strengthening social support systems. The findings of this study underscore the need for the development of policies focused on the mental health of pregnant women. Specifically, if the associations observed in this study reflect causality, enhancing support from families and communities could prevent depression among pregnant women with GDM, as well as among pregnant women in general.

## Supporting information

**S1 Data.**
(XLSX)

## Acknowledgments

We are grateful to two medical facilities, Thai Binh Maternity Hospital and Kim Ngan Clinic in Thai Binh Province and their health staff, and the VALID II research team. Lastly, we thank the research participants for taking part in this study. We also thank Thai Binh health authorities and TBUMP leaders for their support.

## Author Contributions

**Conceptualization:** Ai T. Nguyen, Ib C. Bygbjerg, Ditte S. Linde, Vibeke Rasch.

**Data curation:** Ai T. Nguyen, Hieu Minh Le.

**Formal analysis:** Ai T. Nguyen.

**Funding acquisition:** Tine M. Gammeltoft.

**Investigation:** Ai T. Nguyen.

**Methodology:** Ai T. Nguyen, Dan W. Meyrowitsch, Christina A. Vinter, Ditte S. Linde, Tine M. Gammeltoft, Vibeke Rasch.

**Project administration:** Tine M. Gammeltoft, Vibeke Rasch.

**Resources:** Ai T. Nguyen, Dan W. Meyrowitsch, Ib C. Bygbjerg.

**Software:** Ai T. Nguyen, Hieu Minh Le, Dan W. Meyrowitsch, Jens Søndergaard, Ditte S. Linde.

**Supervision:** Dan W. Meyrowitsch, Jens Søndergaard, Hanh T. T. Nguyen, Christina A. Vinter, Tine M. Gammeltoft, Vibeke Rasch.

**Validation:** Tine M. Gammeltoft, Vibeke Rasch.

**Visualization:** Vibeke Rasch.

**Writing – original draft:** Ai T. Nguyen, Kien Dang Nguyen, Thanh D. Nguyen, Dan W. Meyrowitsch, Ib C. Bygbjerg, Jens Søndergaard, Hanh T. T. Nguyen, Christina A. Vinter, Vibeke Rasch.

**Writing – review & editing:** Ai T. Nguyen, Vibeke Rasch.

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
