## [Decision Letter · Decision Letter 0]

11 Sep 2024

PONE-D-24-32364Social support and symptoms of antenatal depression among women screened for gestational diabetes mellitus: A cross-sectional study in Northern VietnamPLOS ONE

Dear Dr. Nguyen,

Thank you for submitting your manuscript to PLOS ONE. After careful consideration, we feel that it has merit but does not fully meet PLOS ONE’s publication criteria as it currently stands. Therefore, we invite you to submit a revised version of the manuscript that addresses the points raised during the review process.

**I agree with the reviewers that the literature reviewed in the current article is not extensive. The prevalence of GDM among women, particularly in developing and neighboring countries, should be a focal point. Additionally, further studies on social support, GDM, and depression are available and may be reviewed to provide additional insights and emphasize the need for this study.**

**Regarding data collection, who collected the data? Were they qualified and/or trained to assess depression? This should be addressed in the methodology section.**

**Were there any adverse events due to depression? If so, how were they handled?**

**Were women screened for depression prior to recruitment, and were those with depression included in the study?**

**The study reports that 23% of the women were depressed. Was this information shared with the women or their acquaintances? Were these women provided any support by the researchers?**

**For the regression analysis presented in Table 2, what is the reference category for OGTT and physical activity?**

**Lastly, the conclusion should be based on the specific findings of the study. What are the policy implications of this study?**

We look forward to receiving your revised manuscript.

Kind regards,

Ranjan Kumar Prusty, Ph.D.

Academic Editor

PLOS ONE

**Journal Requirements:**

this study was founded by the Danish Ministry of Foreign Affairs.

We are grateful to the Danish Ministry of Foreign Affairs for funding this study, to two medical facilities, Thai Binh Maternity Hospital and Kim Ngan Clinic in Thai Binh Province and their health staff, and the VALID II research team. Lastly, we thank the research participants for taking part in this study. We also thank the Thai Binh health authorities and TBUMP leaders for their support. 

this study was founded by the Danish Ministry of Foreign Affairs

4. We note that there is identifying data in the Supporting Information file "Data_depression". Due to the inclusion of these potentially identifying data, we have removed this file from your file inventory. Prior to sharing human research participant data, authors should consult with an ethics committee to ensure data are shared in accordance with participant consent and all applicable local laws.

-Location data

6. Please amend the manuscript submission data (via Edit Submission) to include author Dr. Hieu Minh Le.

**Additional Editor Comments:**

I agree with the reviewers that the literature reviewed in the current article is not extensive. The prevalence of GDM among women, particularly in developing and neighboring countries, should be a focal point. Additionally, further studies on social support, GDM, and depression are available and may be reviewed to provide additional insights and emphasize the need for this study.

Regarding data collection, who collected the data? Were they qualified and/or trained to assess depression? This should be addressed in the methodology section.

Were there any adverse events due to depression? If so, how were they handled?

Were women screened for depression prior to recruitment, and were those with depression included in the study?

The study reports that 23% of the women were depressed. Was this information shared with the women or their acquaintances? Were these women provided any support by the researchers?

For the regression analysis presented in Table 2, what is the reference category for OGTT and physical activity?

Lastly, the conclusion should be based on the specific findings of the study. What are the policy implications of this study?

Reviewers' comments:

Reviewer's Responses to Questions

**Comments to the Author**

1. Is the manuscript technically sound, and do the data support the conclusions?

Reviewer #1: Yes

Reviewer #2: Yes

2. Has the statistical analysis been performed appropriately and rigorously? 

Reviewer #1: Yes

Reviewer #2: Yes

3. Have the authors made all data underlying the findings in their manuscript fully available?

Reviewer #1: Yes

Reviewer #2: Yes

4. Is the manuscript presented in an intelligible fashion and written in standard English?

Reviewer #1: Yes

Reviewer #2: Yes

5. Review Comments to the Author

**Reviewer #1: **In the abstract:

please provide statistics used. I also suggest to give one more sentence for suggestion for readers. Also, how did you define who have/ dont have depression? For screening tool or diagnosis or else. What does aOR mean? What does CI mean?

In the introduction:

What is the prevalence/incidence of depression among pregnant women and also among GDM? Please review this and write in the introduction.

I think you need to review more about the hormone, diabetes in pregnancy, anatomy and physiology among pregnant women because depression might come from biological factors, hormonal factors, and environmental factors. You wrote only less than one page that could not convince the readers enough. You also talked about stress and depression, which are two concepts, you need to connect that why you should work with the depression instead of stress.

Because you focus on social support, depression, GDM, I suggest you may read both quntitative and qualitative articles. Also, please read more articles related to your research from countries nearby Viet Nam, e.g. feeling and support among pregnant women in Thailand (2020), factors associated with depressive symptoms among women during the antenatal period (2018).

Inclusion criteria might be pregnant women GA 24-28 weeks?

What does Severe Chronic disease covered?

What is the sample size calculation?

Why collected December 2022-Aug 2023?

What is about validity test?

What is the reliability test for SS tool?

How did you do about those having EPDS 10+?

In all tables, if you use abbreviations, please provide full name once!

Some references are old: Ali, 2006, Eaton 1996, etc. I suggest references older than 2015 might be excluded (except theory, tool). I suggest to read some artcle from Thailand, which are close to Viet Nam, such as Phoosuwan wt al, Roomruangwong et al, Pitunupong et al.

**Reviewer #2:** Authors conduct a cross-sectional study of 823 pregnant women screened for gestational diabetes mellitus to assess the association between social support and symptoms of depression. They reported high prevalence of depression symptoms with 23% and a lack of social support associated with increased odds of depression symptoms. The manuscript was well-prepared and I have only minor comments as below.

1. Clarifications are needed for some measurements. For example, authors classified economic status into three categories but didn’t mention their definition.

2. According to table 2, gestational diabetes mellitus is not an associated factor. What is the reasoning to further conduct the stratified analysis by GDM?

6. PLOS authors have the option to publish the peer review history of their article (what does this mean?). If published, this will include your full peer review and any attached files.

Reviewer #1: **Yes: **Assoc.Prof.Dr.Nitikorn Phoosuwan

Reviewer #2: No

---

## [Author Response · Author response to Decision Letter 0]

28 Oct 2024

Dear Ranjan Kumar Prusty, Ph.D.

Thank you very much for your email on the 12nd of September. 

We sincerely appreciate the comments from you and the two reviewers and find that this revision has strengthened the manuscript considerably. Please find our response to the comments from you and the two reviewers point by point.

Query Reviewer: 1 Author’s response

 Abstract: 

1 

please provide statistics used. I also suggest to give one more sentence for suggestion for readers. Also, how did you define who have/ don’t have depression? For screening tool or diagnosis or else. What does aOR mean? What does CI mean? In this manuscript, the Logistic regression analysis was conducted to measure the associations between social support, GDM-related factors, and symptoms of depression. The relationship between social support score and symptoms of depression was evaluated using Spearman’s correlation. 

We used WHO 2013 diagnostic criteria to assess gestational diabetes mellitus, the EPDS tool with a cutoff score of 10 to assess depressive symptoms in pregnant women, and the MSPSS tool to evaluate social support for pregnant women. These issues are clarified in the abstract.

Adjusted OR (aOR) and Confidence interval (CI) are spelled out in the abstract for clarification 

 Introduction: 

2 

What is the prevalence/incidence of depression among pregnant women and also among GDM? Please review this and write in the introduction. The overall prevalence of depression among pregnant women in low- and middle-income countries is 25.5% and in Vietnam 21%. In the GDM group, the prevalence is 24.7% in Thailand, 23.5% in Singapore, 22.5% in Malaysia, and 21.3% in Vietnam. This part of the introduction has been revised in page 4 of manuscript.

3 I think you need to review more about the hormone, diabetes in pregnancy, anatomy and physiology among pregnant women because depression might come from biological factors, hormonal factors, and environmental factors. You wrote only less than one page that could not convince the readers enough. You also talked about stress and depression, which are two concepts, you need to connect that why you should work with the depression instead of stress The revised introduction has more emphasis on hormones, diabetes, and other social factors affecting depression during pregnancy. The revised introduction also explains the rationale for highlighting depression among the mental health factors of pregnant women. Please see more details in page 5.

 Method: 

4 Inclusion criteria might be pregnant women GA 24-28 weeks? Thank you very much for your feedback. In fact, we selected participants from 5-28 weeks (before they come to the hospital for the OGTT) for 2 reasons: 

1. According to our research protocol, participants who agreed to join the study were scheduled for the OGTT in gestational age week 24-28 

2. The questionnaire is quite lengthy, so we divided it into two interview sessions, an inclusion interview and an interview in relation to the OGTT to ensure that pregnant women do not feel it takes too much time, causing discomfort or fatigue.

5 What does Severe Chronic disease covered? It covers acute episodes of chronic diseases, such as kidney disease, heart failure, cancer. This has been added to the paper

6 What is the sample size calculation? This is an explorative descriptive study, thus we did not calculate a sample size but aimed at a sample size of 1000 pregnant women

7 Why collected December 2022-Aug 2023? We started the data collection January 2023 and finalized it August 2023 when we reached the planned sample size. It is explained at page 7.

8 What is about validity test? We conducted a pilot investigation on 30 pregnant women in December. After collecting the survey data, we analyzed and assessed the appropriateness of the toolkit. Subsequently, we made revisions to the sections that were not suitable. 

9 What is the reliability test for SS tool? We used the Cronbach's alpha scale to measure the internal consistency of the toolkit, with a Cronbach's alpha coefficient of 0.91. This has been added to the methodology section of the manuscript in page 8.

10 How did you do about those having EPDS 10+?

 To ensure ethical standards, women who replied “Rarely," "Occasionally," or "Fairly often" when asked about suicidal intentions (Question E10 in EPDS) were referred to a psychologist

11 In all tables, if you use abbreviations, please provide full name once! Explanations for the abbreviations have been listed below each table. 

12 Some references are old: Ali, 2006, Eaton 1996, etc. I suggest references older than 2015 might be excluded (except theory, tool). I suggest to read some artcle from Thailand, which are close to Viet Nam, such as Phoosuwan wt al, Roomruangwong et al, Pitunupong et al. References published before 2015 have been replaced wherever possible and relevant. Hence, we still have some references published before 2015 since they were relevant to the toolkit and theory

 Reviewer #2 

13 Clarifications are needed for some measurements. For example, authors classified economic status into three categories but didn’t mention their definition. The economic status in our study was assessed based on the self-perception and self-reported of the participants. This has been clarified in Table 1.

14 According to table 2, gestational diabetes mellitus is not an associated factor. What is the reasoning to further conduct the stratified analysis by GDM? In our study, we found no association between gestational diabetes and symptoms of depression. However, other studies have indicated a bidirectional relationship between gestational diabetes and depression (as mentioned in the introduction). Additionally, many studies (Friedman LE, et al, Journal of Affective Disorders (2020); Kishore MT, et al, International Journal of Social Psychiatry (2018) and Carroll X, et al, Scientific Reports (2018)) have shown a correlation between social support and depressive symptoms, as well as between social support and gestational diabetes Therefore, we analyzed the relationship between social support and depressive symptoms in two separate groups. As shown in the results of Table 3, the relationship between social support and depressive symptoms in the group of women with gestational diabetes appears to be stronger compared to the group of women without gestational diabetes. This suggests the potential for further research to explore the role of social support and depression among pregnant women in these two groups.

 Editor's Comments to Author 

15 Regarding data collection, who collected the data? Were they qualified and/or trained to assess depression? This should be addressed in the methodology section. In our study, the investigators were nurses/midwives who were thoroughly trained on the research objectives, the research questionnaire, and the approach to participants. This is stated in the revised methodology section (page 7).

16 Were there any adverse events due to depression? If so, how were they handled? We did not encounter any adverse events of women who had high EPDS score. Women who reported any thoughts of suicidal intentions (question E10), were referred for further counseling by a psychologist. 

17 Were women screened for depression prior to recruitment, and were those with depression included in the study? All subjects meeting our research criteria were included in the study. Therefore, pregnant women who had been previously diagnosed with depression could still participate in the study, provided they were not in an acute phase of the illness and able to respond to the research questions. 

18 The study reports that 23% of the women were depressed. Was this information shared with the women or their acquaintances? Were these women provided any support by the researchers? In the study, we only used the scale to detect symptoms of depression, which is referred to as depression screening using the tool. However, we did not conduct a diagnosis of depression (a formal diagnosis of depression must be made by a specialist after the necessary examinations). Since no formal diagnosis of depression was made by a specialist, we did not inform the women or their acquaintances about this status. Pls see more in response to query 16

19 For the regression analysis presented in Table 2, what is the reference category for OGTT and physical activity? In Table 2, this article identifies the mean and standard deviation for the the OGTT test and physical activity and uses the Mann-Whitney U test to determine the relationship between these two variables and depression symptoms.

20 Lastly, the conclusion should be based on the specific findings of the study. What are the policy implications of this study? This has been clarified in the conclusion section of the manuscript in page 22.

---

## [Decision Letter · Decision Letter 1]

14 Nov 2024

Social support and symptoms of antenatal depression among women screened for gestational diabetes mellitus: A cross-sectional study in Northern Vietnam (the VALID II study)

PONE-D-24-32364R1

Dear Dr. Nguyen,

We’re pleased to inform you that your manuscript has been judged scientifically suitable for publication and will be formally accepted for publication once it meets all outstanding technical requirements.

Kind regards,

Ranjan Kumar Prusty, Ph.D.

Academic Editor

PLOS ONE

Additional Editor Comments (optional):

Reviewers' comments:

Reviewer's Responses to Questions

**Comments to the Author**

1. If the authors have adequately addressed your comments raised in a previous round of review and you feel that this manuscript is now acceptable for publication, you may indicate that here to bypass the “Comments to the Author” section, enter your conflict of interest statement in the “Confidential to Editor” section, and submit your "Accept" recommendation.

Reviewer #1: All comments have been addressed

Reviewer #2: All comments have been addressed

2. Is the manuscript technically sound, and do the data support the conclusions?

Reviewer #1: Yes

Reviewer #2: (No Response)

3. Has the statistical analysis been performed appropriately and rigorously? 

Reviewer #1: Yes

Reviewer #2: (No Response)

4. Have the authors made all data underlying the findings in their manuscript fully available?

Reviewer #1: Yes

Reviewer #2: (No Response)

5. Is the manuscript presented in an intelligible fashion and written in standard English?

Reviewer #1: Yes

Reviewer #2: (No Response)

6. Review Comments to the Author

Reviewer #1: Thank you for your revision and your valuable research. I am very satified your work and revision. You have done a good job.

Reviewer #2: (No Response)

7. PLOS authors have the option to publish the peer review history of their article (what does this mean?). If published, this will include your full peer review and any attached files.

Reviewer #1: **Yes: **Nitikorn Phoosuwan

Reviewer #2: No

---

## [Editor Report · Acceptance letter]

26 Nov 2024

PONE-D-24-32364R1 

PLOS ONE

Dear Dr. Nguyen, 

I'm pleased to inform you that your manuscript has been deemed suitable for publication in PLOS ONE. Congratulations! Your manuscript is now being handed over to our production team.

Kind regards, 

on behalf of

Dr. Ranjan Kumar Prusty 

Academic Editor

PLOS ONE